# Flowering Phenology of Selected Elepidote *Rhododendron* L. Taxa

Paweł Maurycy Nowaczyk [1] and Agnieszka Krzymińska-Bródka [2,*]

1. Arboretum Wojsławice, Wroclaw University Botanical Garden, Wojsławice 2, 58-230 Niemcza, Poland
2. Department of Ornamental Plants, Dendrology and Pomology, Poznań University of Life Sciences, Dąbrowskiego 159, 60-637 Poznań, Poland
* Correspondence: agnieszka.krzyminska@up.poznan.pl

**Abstract:** This study aims to evaluate the flowering phenology of selected 95 Elepidote Rhododendron taxa. The flowering period was assessed, distinguishing ten developmental stages. Flowering was observed every three days. The comparative scale prepared indicates the exact succession of bloom of taxa. Flowering of 95 taxa lasted from the end of April to the second half of June. The flowering period of individual cultivars ranged from 2 to 9 weeks. The flowering time of cultivars belonging to the same species was mostly similar. The term of cultivars flowering was in the following tendency order: *R. caucasicum–R. wardii–R. forestii–R. williamsianum–R. catawbiense–R. ponticum.*

**Keywords:** *Rhododendron*; phenology; woody plants

## 1. Introduction

Rhododendrons, members of the *Ericaceae* family, are popular garden shrubs. In Poland, between 1989 and 2000, the number of nurseries specialising in rhododendron production almost doubled. At the same time, there was a fivefold increase in the number of shrubs produced [1]. Ornamental plant nurseries are affiliated with the Polish Nurserymen Association. In 2020, 44 nurseries produced heath plants [2]. Nurseries buy young plants propagated in vitro and grow them until flowering. The value of sales of rhododendrons in Poland in 2000 was close to EUR 100 million [3]. Similarly, in Germany, Winkelmann et al. [4] reported an increase in the number of companies propagating rhododendrons in vitro. In 20 years, the number of laboratories there has almost tripled.

Assortment of *Rhododendron* cultivars is very rich. More than 4100 cultivated rhododendron cultivars are on the Deutsche Genbank Rhododendron checklist [5]. In turn, according to Hoffmann [6], this number exceeds 2000 cultivars per the List of Names of Woody Plants. Marosz [7] reported in 2007 that there were more than 25,000 registered worldwide cultivars.

The available literature does not take into account the vast number of commercially available cultivars. Most information on cultivars can be found in the Deutsche Genbank Rhododendron database and the RHS Rhododendron Checklist with all supplements, as well as Leach [8], Krusmann [9], Czekalski [10], Salle and Greer [11], Schmalscheidt [12–14], Cox [15], Hobbie et al. [16], Grzeszczak-Nowak and Muras [17]. Although many of these were published years ago, they are an essential source of information on rhododendron cultivation. The cultivars recommended by these authors are often no longer in production. Nurseries are promoting new taxa that they consider attractive. Unfortunately, it is often difficult to find information on the cultivars available. There are also no data on the exact time of flowering. Knowing the flowering time is vital to determine suitability for cultivation. This is particularly true of early cultivars whose flowers can be damaged by frost. It is also invaluable to know the succession of bloom of rhododendrons. Selecting cultivars in this manner can help develop a set of plants for green spaces. These sets could provide continuous flowering and long-lasting ornamental quality for at least two months.

This study aims to provide a preliminary evaluation of the flowering phenology of 95 common and lesser-known large-flowered rhododendron taxa, allowing cultivation and use to be determined.

## 2. Materials and Methods

The experiment was carried out from 21 April to 26 June 2020 on the premises of a private heather plant collection located in the southern district of Gdynia at a distance of approximately 1.5 km from the Baltic Sea coastal strip (USDA zone 7A).

The experiment determined the dynamics and flowering phenology of 95 large-flowered rhododendron taxa of the Elepidote group (Table 1). Elepidote is a term that refers to shrubs with leaves without glands on the underside. These usually include shrubs with large evergreen foliage and azaleas with seasonal foliage. The age of the plants studied ranged from 7 to 10 years. All the shrubs observed had reached the generative development stage at least two years earlier.

**Table 1.** Cultivars of rhododendrons used in the experiment, their leading species, parentage and breeder.

| Cultivar | Leading Species | Parentage | Cultivar Breeder, Country |
|---|---|---|---|
| Albert Schweitzer | *R. griffithianum* | unknown | Adriaan van Nes, The Netherlands |
| Album Novum | *R. catawbiense* | hybrid of *R. catawbiense* | Louis van Houtte, Belgium |
| Alfred | *R. catawbiense* | 'Everestianum' F2 | Traugott Jacob Rudolf Seidel, Germany |
| Amaretto | undefined | *Rhododendron dichroanthum* ssp. *scyphocalyx* × 'Marina' | Hans Hachmann, Germany |
| Andantino | *R. strigilosum* | 'Taurus' × 'Carola' | Hans Hachmann, Germany |
| Aprilglocke | *R. williamsianum* | *R. oreodoxa* × *R. williamsianum* | Dietrich Gerhard Hobbie, Germany |
| Arno | *R. catawbiense* | 'Everestianum' × unknown | Traugott Jacob Rudolf Seidel, Germany |
| August Lamken | *R. williamsianum* | 'Doctor V.H. Rutgers' × *R. williamsianum* | Dietrich Gerhard Hobbie, Germany |
| Baden Baden | *R. forestii var. repens* | 'Essex Scarlet' × *R. forrestii* ssp. *forrestii* | Dietrich Gerhard Hobbie, Germany |
| Bananaflip | *R. brachycarpum* | *R. brachycarpum* ssp. *fauriei* × 'Goldsworth Orange' | Hans Hachmann, Germany |
| Bariton | *R. ponticum* | 'Arthur Bedford' × 'Purple Splendour' | Hans Hachmann, Germany |
| Bellini | *R. wardii* | 'Omega' × *R. wardii* | Hans Hachmann, Germany |
| Bengal | *R. forestii var. repens* | 'Essex Scarlet' × *R. forrestii* ssp. *repens* | Dietrich Gerhard Hobbie, Germany |
| Blurettia | *R. yakushimanum* | 'Blue Peter' × 'Koichiro Wada' | Hans Hachmann, Germany |
| Blutopia | *R. catawbiense* | 'Catawbiense Grandiflorum' × 'Arthur Bedford' | Hans Hachmann, Germany |
| Bohlken's Snow Fire | *R. yakushimanum* | 'Koichiro Wada' × 'Hachmann's Diadem' | Heinz Bohlken, Germany |
| Bohlken's Kronjuwel | *R. yakushimanum* | 'Diadem' × 'Germania' | Heinz Bohlken, Germany |
| Buketta | *R. forestii var. repens* | 'Spitfire' × 'Frühlingszauber' | Hans Hachmann, Germany |
| Cassata | *R. catawbiense* | 'Madame Jules Porges' × 'Hachmann's Diadem' | Hans Hachmann, Germany |
| Catawbiense Album | *R. catawbiense* | hybrid of *R. catawbiense* | Waterer, Great Britain |
| Cheer | *R. caucasicum* | 'Cunningham's White' × red flowered *R. catawbiense* | Anthony Shamarello, USA |
| Christmas Cheer | *R. caucasicum* | unknown | unknown |
| Claudine | *R. yakushimanum* | 'Sammetglut' × 'Daisy' | Hans Hachmann, Germany |
| Cosmopolitan | *R. caucasicum* | 'Cunningham's White' × 'Vesuvius' | J. C. Hagen, The Netherlands |
| Cunningham's Blush | *R. caucasicum* | unknown | James Cunningham, Great Britain |
| Cunningham's White | *R. caucasicum* | unknown | J. Cunningham i J. Fraser, Great Britain |
| Danuta | undefined | 'Bellefontaine' × ('Lumina' × 'Perlina') | Hans Hachmann, Germany |
| Dominik | *R. calophytum* | 'Kokardia' × *R. calophytum* | Hans Hachmann, Germany |
| Donar | *R. catawbiense* | 'Mrs Milner' × *R. smirnowii* | Traugott Jacob Rudolf Seidel, Germany |
| Double Dots | undefined | 'Whitney Tiger Lily' × ('Graf Leunknownart' × *R. calophytum*) | Tijs Huismann, The Netherlands |
| Edelweiβ | *R. yakushimanum* | select of *R. degronianum* ssp. *yakushimanum* | Werner Wustemeyer, Johaunknown Wieting, Germany |
| Furnivall's Daughter | *R. griffithianum* | 'Mrs Furnivall' F2 | Anthony Waterer, Great Britain |
| Gartendirektor Rieger | *R. williamsianum* | Adriaan Koster' × *R. williamsianum* | Dietrich Gerhard Hobbie, Germany |
| Gartendirketor Glocker | *R. williamsianum* | 'Doncaster' × *R. williamsianum* | Dietrich Gerhard Hobbie, Germany |

**Table 1.** *Cont.*

| Cultivar | Leading Species | Parentage | Cultivar Breeder, Country |
|---|---|---|---|
| George Sand | *R. wardii* | 'Belkanto' × *R. aureum* | Piotr Muras, Poland |
| Golden Everest | *R. wardii* | 'Bernadette' × 'Goldzauber' | Hans Hachmann, Germany |
| Golden Torch | *R. yakushimanum* | 'Babmi' × (Grosclaude Group × *R. griersonianum*) | Waterer, Great Britain |
| Goldkrone | *R. wardii* | (R.wardii × 'Alice Street') × 'Marina' | Hans Hachmann, Germany |
| Goldprinz | *R. wardii* | 'Festivo' × 'Alice Street' | Hans Hachmann, Germany |
| Golfer | *R. yakushimanum* | *R. degronianum* ssp. *yakushimanum* × *R. pseudochrysanthum* | Warren Berg, USA |
| Gomer Waterer | *R. catawbiense* | Madame Carvalho' × 'Pink Pearl' | Waterer, Great Britain |
| Graf Lennart | *R. wardii* | *R. wardii* × 'Alice Street' | Hans Hachmann, Germany |
| Graziella | *R. roxieanum* | *R. ponticum* × *R. roxieanum* | Hans Hachmann, Germany |
| Hachmann's Charmant | *R. catawbiense* | 'Diadem' × 'Holger' | Hans Hachmann, Germany |
| Hachmann's Corinna | *R. forestii var. repens* | 'Lampion' × ('Ruth Otte' × 'Feuerschein') | Hans Hachmann, Germany |
| Hachmann's Kabarett | undefined | 'Hyperion' × 'Diadem' | Hans Hachmann, Germany |
| Hachmann's Metallica | undefined | 'Marsalla' × 'Peter Alan' | Holger Hachmann, Germany |
| Hachmann's Tanaga | undefined | 'Kabarett' × 'Schneespiegel' | Holger Hachmann, Germany |
| Hachmaunn's Mikado | undefined | ('Lumina' × 'Perlina) × 'Lachsgold' | Hans Hachmann, Germany |
| Halfdan Lem | *R. griffithianum* | 'Jean Marie de Montague' × 'Red Loderi' | Halfdan Lem, USA |
| Helsinki University | *R. brachycarpum* | *R. brachycarpum* | Peter Tigerstedt i Marjatta Uosukainen, Finland |
| Herbstgruβ | undefined | 'Kalamaika' × 'Perlina' | Hans Hachmann, Germany |
| Humboldt | *R. catawbiense* | hybrid of *R. catawbiense* | Traugutt Jacob Rudolf Seidel, Germany |
| Irmelies | *R. williamsianum* | 'Oudijk's Sensation' × 'Marinus Koster' | Hans Hachmann, Germany |
| Jacksonii | *R. caucasicum* | *R. caucasicum* × Nobleanum Group | William Herbert, Great Britain |
| Lee's Dark Purple | *R. catawbiense* | hybrid of *R. catawbiense* | J & C Lee, Great Britain |
| Libretto | *R. ponticum* | 'Lee's Best Purple' × 'Purple Splendour' | Hans Hachmann, Germany |
| Lugano | undefined | 'Fantastica' × ('Ornament' × 'Furnivall's Daughter') | Hans Hachmann, Germany |
| Mars | *R. griffithianum* | hybrid of *R. griffithianum* | Waterer, Great Britain |
| Marylou | *R. wardii* | 'Paprika Spiced' × 'Kalamaika' | Hans Hachmann, Germany |
| Mieszko I | undefined | select of 'Nova Zembla' | Jan Ciepłucha, Poland |
| Millenium Gold | *R. yakushimanum* | 'Goldkrone' × *R. degronianum* ssp. *yakushimanum* | Proefstation vor de Boomkwekerij Boskoop, The Netherlands |
| Mondnacht | *R. campylocarpum* | 'Grünlich' × 'Peucine' | Karl Heinz Hübbers, Germany |
| Morgenrot | *R. yakushimanum* | 'Koichiro Wada' × 'Spitfire' | Hans Hachmann, Germany |
| Motyl | *R. catawbiense* | *R. decorum* × 'Caractacus' | Bohumil Kavka, Czech Republic |
| Nicoletta | *R. yakushimanum* | 'Fantastica × ('Ornament' × 'Furnivall's Daughter') | Hans Hachmann, Germany |
| Nishan | undefined | 'Mondnacht' × 'Kabarett' | Karl Heinz Hübbers, Germany |
| Osmar | *R. williamsianum* | select of *R. williamsianum* | nn, The Netherlands |
| Percy Wisemann | *R. yakushimanum* | F2 from *R. degronianum* ssp. *yakushimanum* × 'Fabia Tangerine' | Waterer, Great Britain |
| Pfauenauge | *R. ponticum* | 'Hyperion' × 'Hachmann's Diadem' | Hans Hachmann, Germany |
| Pohjola's Daughter | *R. brachycarpum* | 'Cunningham's White' × *R. smirnowii* | Marjatta Uosukainen, Finland |
| President Roosevelt | undefined | sport of 'Limbatum' | Mesman, USA |
| Profesor Koziorowski | *R. williamsianum* | select of williamsianum | Antoni Koziorowski, Poland |
| R. scyphocalyx | *R. scyphocalyx* | - | - |
| R. williamsianum | *R. williamsianum* | - | - |
| Rasputin | *R. ponticum* | ('Nova Zemba' × 'Purple Splendour') × 'Purple Splendour' | Hans Hachmann, Germany |
| Robert de Belder | undefined | 'Goldsworth Orange' × *R. dichroanthum subsp. scyphocalyx* | Dietrich Gerhard Hobbie, Germany |
| Rotkäppchen | *R. forestii var. repens* | 'Ruth Otte' × 'Feuerschein' | Hans Hachmann, Germany |
| Royal Butterfly | undefined | 'Haaga' × 'Kabarett' | Jan Ciepłucha, Poland |
| Royal Lilac | undefined | 'Hellikki' × 'Rasputin' | Jan Ciepłucha, Poland |
| Royal Violett | undefined | 'Rasputin' × 'Helsinki University' | Jan Ciepłucha, Poland |

**Table 1.** *Cont.*

| Cultivar | Leading Species | Parentage | Cultivar Breeder, Country |
|---|---|---|---|
| Saffrano | *R. wardii* | unknown | unknown |
| Scarlet Wonder | *R. forestii var. repens* | 'Essex Scarlet' × *R. forrestii* ssp. *forrestii*) | Dietrich Hobbie, Germany |
| September Red | undefined | *R. degronianum* ssp. *yakushimanum* × 'Sammetglut' | Klaus Löptien, Germany |
| Simona | *R. campylocarpum* | 'Harvest Moon' × Letty Edwards Group | Hans Hachmann, Germany |
| Tamarindos | *R. ponticum* | 'Bluebell' × 'Purple Splendour' | Hans Hachmann, Germany |
| Taragona | undefined | 'Double Date' × 'Erato' | Hans Hachmann, Germany |
| Tatjana | *R. yakushimanum* | ('Nova Zembla' × 'Mars') × ('Mars' × 'Koichiro Wada') | Hans Hachmann, Germany |
| Ulrike Jost | undefined | 'Bllefontaine' × ('Lumina' × 'Perlina') | Hans Hachmann, Germany |
| Vater Böhlje | *R. williamsianum* | *R. williamsianum* × 'Catawbiense Compactum' | Dietrich Gerhard Hobbie, Germany |
| Viscy | undefined | 'Diane' × *R. viscidifolium* | Hans Robenek, Germany |
| Vulcan's Flame | *R. griersonianum* | *R. griersonianum* × 'Mars' | Ben Lancaster, USA, |
| Wilgen's Suprise | *R. williamsianum* | Wilgen's Ruby' × *R. williamsianum* | van Wilgen, The Netherlands |
| Wine & Roses | *R. nerifolium* | unknown | Kenneth Cox, Great Britain |
| XXL | undefined | unknown | Johan Vandergaegen, Belgium |

Plants were grown in the ground, under canopy conditions of a Class 5 pine forest. The cultivation substrate was a mixture of acidic peat with a high pH of 4.5–5.5 and native soil (sand with clay subsoil and compost soil) at an estimated volume ratio of 3:1:2. The soil surface was mulched with composted pine bark.

No plants were watered, and no chemical protection treatments were applied during the experiment. Plants grew in conditions as close to natural as possible.

The different cultivars of shrubs were monitored for bud development. Subsequent stages were determined using a developed 10-degree comparative scale (Table 2). Stages 1–3 defined the development of the inflorescence bud. Stage 4 was established when the first flower in the inflorescence had developed. Stage 5 corresponded to the development of at least half of the flowers in the inflorescence. Full bloom (all flowers open) was designated as Stage 6. Stages 7, 8 and 9 described shedding blossom. Stage 7 determined the first wilted flowers, Stage 8 determined that all flowers wilted, and Stage 9 determined flower browning and dropping off. Reaching Stage 10 was indicative of the completely finished flowering of the shrub (no flower petals).

Damage caused by spring frost and heavy rain was also reported. Observations were made every three days. During the experiment, no patchy development of individual inflorescences was observed on the shrubs.

Based on Muras [18], this study used the Körung method. It relies on the cross-assessment of single, rare specimens against individual traits. This work studied flowering dynamics.

Figure 1 shows meteorological data from the Institute of Meteorology and Water Management from the Gdańsk-Rębiechowo station (10 km in a straight line from the collection site). The data include maximum temperature, minimum temperature, average temperature and amount of rainfall.

**Table 2.** Flower development scale used in the work.

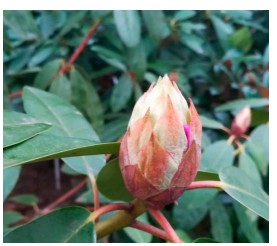

Stage 1. Inflorescence bud lax, petal colour visible

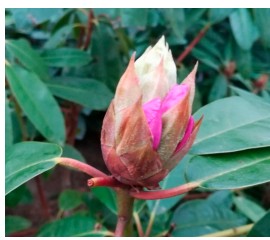

Stage 2. First flower petals visible in buds

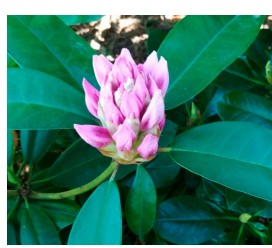

Stage 3. Flower buds visible, undeveloped, no visible bud scales

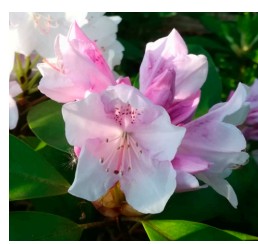

Stage 4. First open flower in the inflorescence

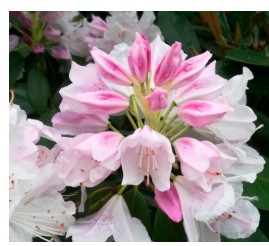

Stage 5. Half of the flowers in the inflorescence open

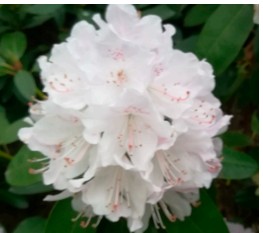

Stage 6. All flowers open

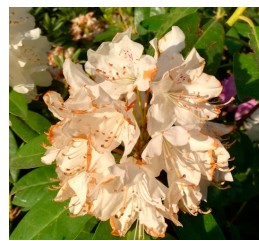

Stage 7. First wilted flowers

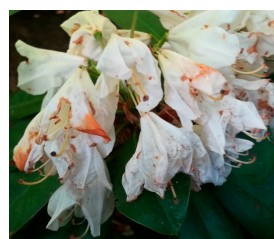

Stage 8. All flowers wilted

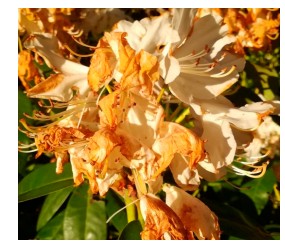

Stage 9. Flowers browning and dropping off

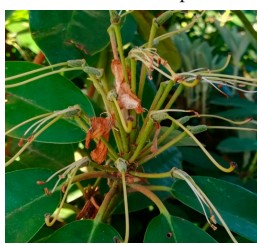

Stage 10. No flower petals

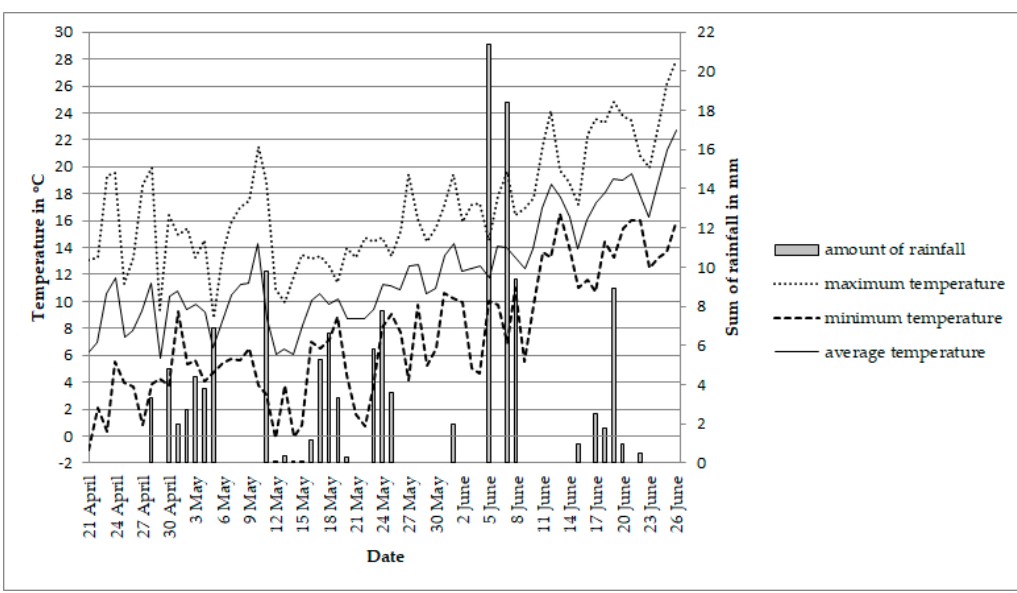

**Figure 1.** Meteorological data obtained within the period of observation in 2020.

### 3. Results and Discussion

The 95 rhododendron taxa assessed differed in the timing of the appearance of the different flowering stages (Table 3). The beginning of inflorescence bud development of the first cultivars occurred in the 17th week of the year. Five—'Andantino', 'Jacksonii', 'Cheer', 'Christmas Cheer' and 'George Sand'—reached their first stage of development on 21 April 2020. The latest started to develop in late May and early June. The difference between the start of flowering in the first variety, 'Cheer', and the wilting of the flowers in the last, 'Rasputin', was 45 days.

The experiment showed that the largest number of cultivars was in full bloom in the 23rd week of the year. In this week, 23% of all observed cultivars developed. However, the first cultivars were flowering as early as week 18 (7%) and the last cultivars at week 24 (6%). The shortest flowering was found in 'Cheer' and its sister variety 'Christmas Cheer'. 'Cunningham's White', on the other hand, retained the longest ornamentation. The long flowering period of this variety is in line with the data presented by Malciute et al. [19]. In the Lithuanian climate, flowering time was 26–29 days, while in our experiment, it was about three weeks. According to the cited authors, the cultivars 'Goldkrone' and 'Bengal' show short flowering (9–15 days). The flowering of the first variety under Polish conditions lasted less than two weeks, while 'Bengal' remained ornamental for about 20 days. Under both Polish and Lithuanian conditions, flowering lasting 16–25 days was found in 'Blurettia', 'Gartendirektor Rieger', 'Hachmann's Charmant', 'Helsinki University', 'Pohjola's Daughter' and 'Rasputin'. However, according to our research, 'Album Novum', 'Baden-Baden' and 'Percy Wiseman' remained ornamental for a shorter period than reported by Malciute et al. [19]. Most of the shrubs observed presented a flowering time of approximately 1.5 weeks. As reported by Sauliene et al. [20], more than one-third of rhododendron shrub cultivars have two-week flowering, and almost 5% maintain their inflorescences for more than three weeks. The vast majority of shrubs (42%) flowered for two to three weeks. In their opinion, however, the intensity of flowering may depend on the degree of acclimatisation and proper nutrition of the particular specimen of the cultivars studied.

**Table 3.** Flowering phenology of selected 95 *Rhododendron* taxa.

| Month | April | | | | May | | | | | | | | | | June | | | | | | | |
|---|---|---|---|---|---|---|---|---|---|---|---|---|---|---|---|---|---|---|---|---|---|---|
| Day of Month | 21 | 24 | 27 | 30 | 3 | 6 | 9 | 12 | 15 | 18 | 21 | 24 | 27 | 30 | 2 | 5 | 8 | 11 | 14 | 17 | 20 | 23 |
| Week of the Year | 17 | | | 18 | | | 19 | | | 20 | | | 21 | | | 22 | | 23 | | 24 | | 25 | 26 |
| STR *—Andantino | 1 ** | 1/2 | 3 | 4 | 6 | 6 | 7 | 7/8 | 9/10 | | | | | | | | | | | | | |
| CAU—Jacksonii | 1 | 2 | 3 | 4/5 | 6 | 7 | 8 | 9 | 10 | | | | | | | | | | | | | |
| CAU—Cheer | 1/2 | 3 | 4 | 5/6 | 7 | 8 | 10 | | | | | | | | | | | | | | | |
| CAU—Christmas Cheer | 2 | 2 | 3 | 4 | 5(F) | 7 | 10 | | | | | | | | | | | | | | | |
| WAR—George Sand | 1/2 | 3 | 5 | 6 | 7 | 9 | 9/10 | 10 | | | | | | | | | | | | | | |
| FOR—Bengal | | 1 | 2 | 3 | 5 | 6 | 6 | 6 | 6 | 6/7 | 6/7 | 6/7 | 7 | 7 | 7/8 | 10 | | | | | | |
| FOR—Buketta | | 1 | 3 | 5 | 6 | 6 | 7 | 7/8 | 8/9 | 9 | 9/10 | 10 | | | | | | | | | | |
| CAU—Cosmopolitan | | 1 | 2 | 3 | 5 | 6 | 6 | 6 | 6 | 6/7 | 7 | 8 | 8 | 9 | 9 | 10 | | | | | | |
| WAR—Graf Lennart | | 1 | 2/3 | 4 | 5 | 6 | 6 | 7 | 7/8 | 10 | | | | | | | | | | | | |
| UND—Herbstgruβ | | 1/2 | 3 | 4 | 5 | 6 | 6 | 6 | 7 | 7 | 8 | 8 | 9 | 10 | | | | | | | | |
| BRA—Pohjola's Daughter | | 1 | 2 | 3 | 3 | 4 | 5 | 6 | 6 | 7 | 7 | 7 | 7/8 | 8 | 9/10 | 10 | | | | | | |
| FOR—Scarlet Wonder | | 1 | 2 | 4 | 6 | 6 | 6 | 6 | 6/7 | 7 | 7/8 | 8 | 8 | 9 | 10 | | | | | | | |
| CAT—Arno | | | 1 | 2 | 3 | 4 | 5/6 | 6 | 6 | 6 | 6 | 67 | 7 | 7 | 7/8 | 7/8 | 8/9 | 8/9 | 9/10 | 10 | | |
| FOR—Baden Baden | | | 1 | 2 | 3 | 5 | 6 | 6 | 6 | 7(R) | 7/8 | 10 | | | | | | | | | | |
| CAU—Cunningham's Blush | | | 1/2 | 3 | 4 | 5 | 6 | 6/7 | 6/7 | 7 | 9/10 | 10 | | | | | | | | | | |
| CAU—Cunningham's White | | | 1 | 2 | 3 | 4 | 5 | 5/6 | 6 | 6 | 6 | 6 | 6 | 6/7 | 7/8 | 9/10 | 10 | | | | | |
| WAR—Goldkrone | | | 2 | 3 | 5 | 5 | 6 | 6/7 | 6/7 | 7 | 8 | 8 | 8/9 | 9 | 9/10 | 10 | | | | | | |
| WAR—Goldprinz | | | 1 | 3 | 5 | 6 | 6 | 6 | 6 | 6/7 | 6/7 | 7/8 | 8 | 8 | 8/9 | 9/10 | 10 | | | | | |
| UND—President Roosevelt | | | 2/3 | 4 | 5 | 7 | 7 | 8 | 8/9 | 8/9 | 9 | 9/10 | 10 | | | | | | | | | |
| FOR—Rotkäppchen | | | 1/2 | 3 | 4 | 6 | 6 | 6/7 | 6/7 | 6/7 | 7/8 | 10 | | | | | | | | | | |
| WIL—R. williamsianum | | | 1 | 2/3 | 3 (F) | 3 | 4 | 4 | 4 | 7/8 | 8/9 | 8/9 | 9/10 | | | | | | | | | |
| WIL—Aprilglocke | | | | 1 | 2 | 2 | 4 | 5 | 6 | 6 | 7/8 | 8 | 8/9 | 9 | 9/10 | 10 | | | | | | |
| WIL—August Lamken | | | | 1 | 2 | 3 | 4 | 6 | 6(F) | 6 | 7/8 | 7/8 | 8 | 8 | 8/9 | 10 | | | | | | |
| CAT—Catawbiense Album | | | | 1/2 | 2 | 3 | 4 | 4 | 5 | 6 | 6 | 6/7 | 6/7 | 7 | 7/8 | 9/10 | 10 | | | | | |
| FOR—Hachmann's Corinna | | | | 1 | 2 | 3 | 5 | 6 | 6 | 6 | 6/7 | 7/8 | 8/9 | 10 | | | | | | | | |
| CAL—Dominik | | | | 1 | 2 | 4 | 6 | 6/7 | 7 | 7 | 8/9 | 9/10 | 9/10 | 10 | | | | | | | | |
| UND—Double Dots | | | | 1 | 1 | 3 | 5 | 6(F) | 7 | 7 | 7 | 7 | 8 | 8 | 8/9 | 9/10 | 10 | | | | | |
| WIL—Gartendirektor Rieger | | | | 1 | 3 | 4 | 5 | 5 | 6 | 6 | 7/8 | 8 | 8/9 | 9 | 10 | | | | | | | |
| WIL—Gartendirketor Glocker | | | | 1 | 2 | 3 | 5 | 5 | 6 | 7(R) | 7/8 | 8/9 | 8/9 | 9 | 9 | 10 | | | | | | |
| WIL—Irmelies | | | | 1 | 2 | 3 | 4 | 6 | 6/7 | 7 | 7 | 8 | 8 | 9/10 | 10 | | | | | | | |
| YAK—Nicoletta | | | | 1 | 3 | 4 | 5 | 6 | 6 | 6 | 6/7 | 7 | 8 | 8/9 | 8/9 | 9/10 | 10 | | | | | |
| WIL—Profesor Koziorowski | | | | 1 | 2 | 3 | 4 | 6 | 6 | 6/7 | 6/7 | 7 | 8 | 9 | 10 | | | | | | | |
| WAR—Saffrano | | | | 1 | 2 | 3 | 5 | 5 | 6 | 7 | 8 | 9/10 | 10 | | | | | | | | | |
| CAM—Simona | | | | 1 | 3 | 4 | 6 | 6 | 6 | 6 | 7 | 7/8 | 8/9 | 9/10 | | | | | | | | |
| WIL—Wilgen's Suprise | | | | 1 | 2 | 4 | 6 | 6 | 6/7 | 6/7 | 6/7 | 7 | 7/8 | 9 | 10 | | | | | | | |
| NER—Wine & Roses | | | | 1 | 2 | 3 | 5 | 5(F) | 5/6 | 7 | 8 | 8/9 | 9/10 | | | | | | | | | |
| UND—XXL | | | | 1/2 | 3 | 4 | 6 | 6 | 6 | 6/7 | 7 | 8 | 8/9 | 8/9 | 8/9 | 9/10 | | | | | | |
| YAK—Bohlken's Snow Fire | | | | | 1 | 2 | 3 | 4 | 5 | 6 | 6 | 6 | 7 | 7/8 | 8 | 9 | 10 | | | | | |
| YAK—Golfer | | | | | 1 | 3 | 5 | 5 | 6 | 7 | 7 | 9 | 9/10 | 10 | | | | | | | | |
| YAK—Millenium Gold | | | | | 1 | 2 | 4 | 5 | 6 | 6/7 | 6/7 | 8/9 | 8/9 | 9 | 9/10 | 10 | | | | | | |
| UND—Robert de Belder | | | | | 1 | 2 | 4 | 4/5 | 6 | 6 | 6/7 | 7/8 | 8/9 | 10 | | | | | | | | |
| WIL—Vater Böhlje | | | | | 1 | 2 | 4 | 4 | 5 | 6(R) | 7 | 8 | 8 | 8/9 | 9 | 9/10 | | | | | | |
| YAK—Bohlken's Kronjuwel | | | | | | 1 | 2 | 3 | 3 | 5 | 6 | 6 | 6 | 7 | 7/8 | 9/10 | | | | | | |
| GRI—Furnivall's Daughter | | | | | | 1 | 2 | 4 | 5 | 6 | 6 | 6 | 6 | 6 | 6/7 | 7 | 8/9 | 10 | | | | |
| ROX—Graziella | | | | | | 1 | 2 | 3 | 3 | 4 | 5/6 | 5/6 | 7 | 7 | 7/8 | 8/9 | 9/10 | 10 | | | | |
| GRI—Halfdan Lem | | | | | | 1 | 2 | 3 | 4 | 5 | 5/6 | 6 | 6/7 | 8 | 8/9 | 10 | | | | | | |

**Table 3.** *Cont.*

| Month | April | | | | May | | | | | | | | | | June | | | | | | | |
|---|---|---|---|---|---|---|---|---|---|---|---|---|---|---|---|---|---|---|---|---|---|---|
| **Day of Month** | 21 | 24 | 27 | 30 | 3 | 6 | 9 | 12 | 15 | 18 | 21 | 24 | 27 | 30 | 2 | 5 | 8 | 11 | 14 | 17 | 20 | 23 |
| **Week of the Year** | | 17 | | 18 | | | 19 | | 20 | | 21 | | 22 | | | 23 | | 24 | | 25 | | 26 |
| CAT—Motyl | | | | | | | 1 | 2/3 | 3/4 | 5/6 | 6 | 6 | 6 | 7 | 7 | 8 | 8/9 | 10 | | | | |
| WIL—Osmar | | | | | | | 1 | 3 | 3/4 | 3/4 | 4(F) | 6/7 | 7 | 7/8 | 8 | 8/9 | 9/10 | 10 | | | | |
| UND—Viscy | | | | | | | 1 | 2 | 3 | 4 | 5 | 5/6 | 5/6 | 5/6 | 6 | 6 | 6/7 | 7/8 | 10 | | | |
| CAT—Blutopia | | | | | | | | 1 | 1/2 | 2 | 3 | 5 | 5 | 6 | 6 | 6 | 6 | 7(R) | 7/8 | 8/9 | 10 | |
| CAT—Cassata | | | | | | | | 1 | 1 | 1 | 2 | 2/3 | 3 | 3/4 | 5 | 6 | 6 | 6 | 6 | 7 | 8 | 10 |
| WAR—Golden Everest | | | | | | | 1 | 2 | 2 | 3 | 3 | 4 | 5 | 5/6 | 6 | 6 | 6 | 6 | 7 | 7 | 8 | 9/10 |
| WAR—Marylou | | | | | | | | 1 | 1/2 | 3 | 4 | 5/6 | 6 | 6 | 7 | 7/8 | 9/10 | 10 | | | | |
| UND—Nishan | | | | | | | | 1 | 2/3 | 3 | 4 | 4/5 | 6 | 6 | 7 | 8 | 10 | | | | | |
| UND—Danuta | | | | | | | 1/2 | 3 | 4 | 6 | 6 | 6 | 6 | 6 | 6/7 | 9/10 | 10 | | | | | |
| BRA—Bananaflip | | | | | | | | 1 | 3 | 3/4 | 4/5 | 5 | 6 | 6/7 | 7/8 | 9/10 | 10 | | | | | |
| WAR—Bellini | | | | | | | | 1 | 2 | 2/3 | 4 | 4/5 | 6 | 6 | 6 | 7/8 | 8/9 | 10 | | | | |
| UND—Ulrike Jost | | | | | | | | 1 | 1/2 | 3 | 4 | 5 | 6 | 6 | 6/7 | 7 | 7/8 | 9/10 | | | | |
| YAK—Tatjana | | | | | | | | 1 | 2/3 | 3 | 4/5 | 6 | 6 | 6 | 6 | 8 | 10(R) | | | | | |
| YAK—Blurettia | | | | | | | | 1 | 2/3 | 4 | 5 | 6 | 6 | 6 | 6/7 | 7 | 7/8 | 8 | 8/9 | 9 | 10 | |
| YAK—Golden Torch | | | | | | | | 1 | 2/3 | 3/4 | 4/5 | 5/6 | 6 | 6 | 6/7 | 6/7 | 7 | 7/8 | 8 | 8/9 | 10 | |
| UND—Hachmann's Mikado | | | | | | | | | | 1 | 2 | 3 | 3/4 | 3/4 | 5 | 5/6 | 6 | 6 | 6/7 | 8/9 | 9/10 | |
| YAK—Percy Wisemann | | | | | | | | | | 1 | 2 | 2/3 | 4 | 6 | 6 | 6/7 | 6/7 | 10(R) | | | | |
| UND—Royal Butterfly | | | | | | | | | | 1 | 2 | 2/3 | 4/5 | 6 | 6 | 6 | 7 | 9/10 | 9/10 | 10 | | |
| UND—Hachmann's Kabarett | | | | | | | | | | 1 | 2/3 | 3 | 4 | 6 | 6 | 6 | 7 | 9/10 | 10 | | | |
| SCY—scyphocalyx | | | | | | | | | | 1 | 2/3 | 3/4 | 4/5 | 6 | 6 | 6/7 | 7/8 | 7/8 | 8 | 9/10 | | |
| UND—September Red | | | | | | | | | | | 1 | 2 | 3 | 4 | 5 | 6 | 7 | 8(R) | 8 | 8/9 | 10 | |
| CAT—Gomer Waterer | | | | | | | | | | | 1 | 1/2 | 2/3 | 3 | 4 | 5 | 6 | 6 | 6/7 | 8 | 10 | |
| CAT—Humboldt | | | | | | | | | | | 1 | 2 | 3 | 3/4 | 5/6 | 6 | 6 | 6 | 6 | 7 | 9/10 | 10 |
| YAK—Claudine | | | | | | | | | | | 1 | 2 | 3/4 | 4 | 5 | 6 | 6 | 6/7 | 6/7 | 8 | 9/10 | 10 |
| YAK—Morgenrot | | | | | | | | | | | 1 | 2 | 3/4 | 5/6 | 6 | 6 | 6/7 | 9/10 | 10 | | | |
| CAM—Mondnacht | | | | | | | | | | | 1 | 2/3 | 3/4 | 3/4 | 5 | 6 | 6 | 7 | 7 | 8/9 | 10 | |
| CAT—Hachmann's Charmant | | | | | | | | | | | 1/2 | 3 | 3 | 3 | 5 | 6 | 6 | 6 | 6/7 | 7/8 | 10 | |
| UND—Lugano | | | | | | | | | | | 1/2 | 3 | 3/4 | 5 | 6 | 6 | 6 | 6 | 6 | 8 | 9/10 | 10 |
| UND—Vulcan's Flame | | | | | | | | | | | 1/2 | 2/3 | 3/4 | 4 | 5 | 6 | 6 | 6 | 6 | 8 | 10 | |
| GRI—Albert Schweitzer | | | | | | | | | | | 1/2 | 2/3 | 3/4 | 4 | 5 | 6 | 6/7 | 8 | 8/9 | 10/9 | | |
| CAT—Alfred | | | | | | | | | | | | 1 | 2 | 3 | 5 | 6 | 6 | 6/7 | 8 | 9/10 | | |
| CAT—Album Novum | | | | | | | | | | | | 1 | 2 | 2/3 | 3 | 5 | 6 | 6/7 | 8 | 8/9 | 10 | |
| UND—Royal Violett | | | | | | | | | | | | 1 | 2 | 2/3 | 5/6 | 6 | 6/7 | 6/7 | 7/8 | 8/9 | 10 | |
| PON—Tamarindos | | | | | | | | | | | | 1 | 2/3 | 3 | 5 | 6 | 6 | 6 | 6 | 7 | 9/10 | |
| BRA—Helsinki University | | | | | | | | | | | | 1 | 2/3 | 4 | 5 | 6 | 6 | 7 | 8 | 9 | 10 | |
| UND—Hachmann's Tanaga | | | | | | | | | | | | 1 | 2/3 | 4 | 5/6 | 6 | 6/7 | 7 | 7/8 | 9/10 | | |
| UND—Amaretto | | | | | | | | | | | | 1 | 3/4 | 4 | 4/5 | 5 | 6 | 6 | 6/7 | 7/8 | 8 | 10 |
| CAT—Donar | | | | | | | | | | | | 2 | 3 | 4 | 6 | 6 | 6 | 6/7 | 7/8 | 8 | 9/10 | 10 |
| UND—Taragona | | | | | | | | | | | | 1/2 | 2 | 2/3 | 3 | 4/5 | 5/6 | 6 | 6 | 7/8 | 9/10 | |
| UND—Hachmann's Metallica | | | | | | | | | | | | 1/2 | 3 | 4/5 | 6 | 6 | 7 | 7/8 | 8 | 9 | 10 | |
| UND—Mieszko I | | | | | | | | | | | | 1/2 | 3/4 | 4 | 5 | 6 | 7 | 7/8 | 8 | 9 | 10 | |
| UND—Royal Lilac | | | | | | | | | | | | | 1 | 2 | 2/3 | 4 | 6 | 6 | 6 | 7 | 8/9 | 9/10 |
| CAT—Lee's Dark Purple | | | | | | | | | | | | | 1 | 2/3 | 3/4 | 4 | 4 | 4/5 | 6 | 6 | 8/9 | 10 |
| YAK—Edelweiß | | | | | | | | | | | | | 2/3 | 2/3 | 4 | 6 | 7 | 7(R) | 7/8 | 9/10 | 10 | |
| GRI—Mars | | | | | | | | | | | | | | 1 | 2/3 | 4 | 6 | 6 | 6 | 7/8 | 10 | |

**Table 3.** *Cont.*

| Month | April | | | | May | | | | | | | | | | June | | | | | | | |
|---|---|---|---|---|---|---|---|---|---|---|---|---|---|---|---|---|---|---|---|---|---|---|
| **Day of Month** | 21 | 24 | 27 | 30 | 3 | 6 | 9 | 12 | 15 | 18 | 21 | 24 | 27 | 30 | 2 | 5 | 8 | 11 | 14 | 17 | 20 | 23 |
| **Week of the Year** | | 17 | | | 18 | | | 19 | | | 20 | | 21 | | 22 | | 23 | | 24 | | 25 | 26 |
| PON—Rasputin | | | | | | | | | | | | | 1/2 | 2/3 | 4 | 5/6 | 6 | 6 | 6 | 7/8 | 9/10 | |
| PON—Pfauenauge | | | | | | | | | | | | | 1/2 | 3 | 3/4 | 4/5 | 6 | 6 | 6/7 | 8/9 | 10 | |
| PON—Libretto | | | | | | | | | | | | | | 1 | 2/3 | 3 | 3/4 | 6 | 6 | 7/8 | 10 | |
| PON—Bariton | | | | | | | | | | | | | | 1/2 | 2 | 3/4 | 4/5 | 6 | 6/7 | 7/8 | 10 | |

* Leading species–BRA-*R. brachycarpum*; CAL-*R. calophytum*; CAM-*R. campylocarpum*; CAU-*R. caucasicum*; CAT-*R. catawbiense*; FOR-*R. forestii var. repens*; GRE-*R. griersonianum*; GRI-*R. griffithianum*; NER-*R. nerifolium*; PON-*R. ponticum*; ROX-*R. roxieanum*; SCY-*R. scyphocalyx*; STR-*R. strigilosum*; UND–undefined; WAR-*R. wardii*; WIL-*R. williamsianum*; YAK-*R. yakushimanum*; ** Stage 1. Inflorescence bud lax, petal colour visible; Stage 2. First flower petals visible in buds; Stage 3. Flower buds visible, undeveloped, no visible bud scales; Stage 4. First open flower in the inflorescence; Stage 5. Half of the flowers in the inflorescence open; Stage 6. All flowers open; Stage 7. First wilted flowers; Stage 8. All flowers wilted; Stage 9. Flower browning and dropping off; Stage 10. No flower petals.

Muras [21] reported that the flowering of 'Baden-Baden' occurs from the beginning to the second half of May, while 'Jacksonii' flowered from the third week of April to the first week of May. These findings are confirmed by our research. Flowering of *Rhododendron brachycarpum* occurs from late May to mid-June, depending on the geographical source of the specimen. In our experiment, the flowering of 'Helsinki University', a relative of this species, occurred from mid-June onwards. 'Pohjola's Daughter' was an earlier variety of *R. brachycarpum*, flowering from the 20th week of the year. Flowering of *R. degronianum* subsp. *yakushimanum* lasts from the beginning to almost the end of May. Similar results were found in almost all tested cultivars of this species in our experiment. The exception was 'Edelweiβ', which started flowering in early June. It should be noted that it also has exceptionally ornamental leaves. The upper side of the leaf blade is covered with silvery-white tomentum. The underside of the leaves has a brown lining. It is one of the best cultivars in terms of ornamental foliage. Under British conditions, it flowers from early to mid-May. Leaf blades with multicoloured features, modified structure or mossiness are also found in other cultivars examined, including 'Andantino', 'Dominik', 'Golfer', 'Graziella', 'President Roosevelt' and 'Wine & Roses'. Growing them in the garden offers the possibility of a year-round ornamental effect.

Aleksandrova [22] listed five groups of cultivars depending on flowering time. She identified five categories of cultivars: very early, medium-early, early, late and very late. The proposed classification was used by Malciute et al. [19]. 'Helsinki University', 'Blurettia' and 'Album Novum' have been classified as early cultivars (flowering from day 145 to day 159 of the year), even though they are considered medium-late and late cultivars. Hobbie et al. [16] described 'Album Novum' as a late variety and 'Blurettia' as a late April/May flowering variety. 'Hachmann's Charmant' and 'Percy Wiseman' start flowering in Poland in the second half of May. Malciute et al. [19] considered them to be late cultivars.

Caprar et al. [23], conducting observations of rhododendrons at the Jibou Botanical Garden in Romania, divided them into seven groups: from very early and early, through intermediate, early intermediate and late intermediate to late and very late. The variety 'Scarlet Wonder' was described as intermediate. The term is comparable to that found in our work.

By creating groups that take into account the earliness of flowering, a link to pedigree can be established. The earliest flowering cultivars in our experiment were developed with a distant contribution from *R. caucasicum*. The cultivars 'Bengal' and those derived from *R. forrestii* started flowering almost simultaneously in early May (18th/19th week). Yellow-flowered R. wardii hybrids also started to develop inflorescences in the first days of May (18th/19th week). The cultivar that deviated from this date was 'Bellini', which flowered in the second half of May (21st week). The *R. williamsianum* hybrid group was distinguished by flowering from week 20 onwards; the latest cultivar was 'Osmar', flowering at week 21. Most rhododendrons flowering from week 20 onwards included cultivars not assigned to any group. *R. catawbiense* hybrids: 'Blutopia', 'Gomer Waterer', 'Humboldt', 'Alfred', 'Album Novum' and 'Mars' showed flower development from weeks 22 and 23. The latest cultivars were purple-flowered hybrids developed from *R. ponticum*. The flowering of *R. yakushimanum* cultivars was very diverse and started from the 19th to 22nd week. Such exact information on flowering is not provided on the labels when the plants are purchased. The timing is often averaged out as late May and early June.

Blatsios [24] selected four groups of rhododendron shrubs grown from free pollination. Most of the shrubs (55%) flowered in mid-May. Very early flowering occurred in 43% of the shrubs, and early flowering in 23%. Late flowering was reported in 19%, and very late flowering in 1% of the hybrids produced.

Malciute et al. [19] found the Humboldt variety to be long flowering. However, under Polish cultivation conditions in the coastal zone in our study, its flowering lasted about 1.5 weeks. It is suggested that this may have been a result, among other things, of water deficiency in the soil during inflorescence development.

The effect of the amount of precipitation on inflorescence longevity was also observed during the experiment. Cultivars starting to flower in 22nd week and in 24th week had lower inflorescence longevity than those flowering during heavy rainfall.

Currently, growing new cultivars aims to extend the flowering period. Increasing attention is being paid to cultivars flowering from early June onwards, e.g., 'Leo' or 'Albarello' [25]. In recent years, cultivars showing partial flowering in autumn have also been developed: 'Herbstfreude', 'Herbstzauber' and 'September Charm' [26]. In addition, partial autumn flowering of the cultivars 'Weinlese' and 'Karminkissen' is also observed in the research garden. Repeat flowering in autumn is also characteristic of the two cultivars evaluated in the experiment, i.e., 'Herbstgruβ' and 'September Red'. Partial repeat flowering also occurs in 'Cunningham's White' and 'Cheer', sometimes lasting until the first frost. Malciute et al. [19] confirmed that 'Cunningham's White' blooms twice a year. The same applies to 'Pohjola's Daughter' and 'Nova Zembla'. In the last one, the autumn flowering period was longer than in the spring one. Ciepłucha [27] described the partial autumn flowering of 'Królowa Jadwiga'. In the online promotional material of this author's nursery, there is also information about the autumn flowering of the 'Lugano' and 'Bezděz' cultivars.

In addition to the main flowering period of large-flowered rhododendrons, summer also sees the flowering of other heath shrubs, which can extend the ornamental appeal of the setting created. A less common group includes Japanese azalea hybrids developed using *Rhododendron nakaharae*. The cultivars 'Juliette', 'Hachmann's Evita', 'Michael Hill' and 'Late Love' have red and pink flowers, developing from mid-June. Additionally, azalea shrubs with seasonal foliage developed using *R. viscosum* bloom in June and July. They also have highly fragrant flowers. The earliest hybrids are *R. dauricum* and *R. carolinianum* 'Stacatto', 'April Gem', 'April Rose', and five cultivars derived from the P.J.M. group [20]. Along with these, *R. mucronulatum* blooms eight days earlier than *R. dauricum* [28]. Therefore, when deciding which cultivars to choose, it is worth knowing the exact flowering time to achieve as long a flowering succession as possible.

Our work did not consider the flowering of the three most common cultivars, i.e., 'Nova Zembla', 'Roseum Elegans' and 'Catawbiense Grandiflorum'. These cultivars are the most commonly available commercially. However, they were not included, due to their incorporation into the observations of cultivars that are less widespread or practically unknown and yet worth propagating. Due to the extensive area of this research topic, many other shrubs were not included: the most commonly offered evergreen rhododendrons, large-flowered azaleas, 'Japanese' azaleas and dwarf rhododendrons. However, this gap should not be seen as negative. The authors' collection, which has been amassed and expanded annually with new taxa, provides the opportunity to continue research in the years to come. It will make it possible to better understand the flowering phenology of the various representatives of the genus *Rhododendron*.

Based on these observations, rhododendrons could be selected for gardens. For small gardens, we can recommended rhododendrons with white and pink flowers: 'Cheer', 'Herbstgruss', 'Nicoletta', 'Butterfly', 'Danuta', 'Claudine' and 'Edelweiss', or with red flowers: 'Bengal', 'Halfdan Lem', 'September Red' and 'Taragona'. For small gardens, we can also recommend 'George Sand', 'Scarlet Wonder', 'August Lamken', 'Osmar', 'Golden Torch' and 'Lugano'.

**Author Contributions:** Conceptualization, P.M.N. and A.K.-B.; methodology, P.M.N. and A.K.-B.; software, P.M.N. and A.K.-B.; validation, P.M.N. and A.K.-B.; formal analysis, P.M.N. and A.K.-B.; investigation, P.M.N. and A.K.-B.; resources, P.M.N. and A.K.-B.; data curation, P.M.N. and A.K.-B.; writing—original draft preparation, P.M.N. and A.K.-B.; writing—review and editing, A.K.-B.; visualization, P.M.N. and A.K.-B.; supervision, A.K.-B.; funding acquisition, A.K.-B. All authors have read and agreed to the published version of the manuscript.

**Funding:** The publication was co-financed within the framework of the Polish Ministry of Science and Higher Education's program: "Regional Initiative Excelence" in the years 2019–2023 (No. 005/RID/2018/19), financing amount 12,200,000 PLN.

**Institutional Review Board Statement:** Not applicable.

**Data Availability Statement:** Not applicable.

**Conflicts of Interest:** The authors declare no conflict of interest.

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
