# Peer review of "Flowering Phenology of Selected Elepidote Rhododendron L. Taxa"

_agriculture, doi:10.3390/agriculture13010036_

Round 1

Reviewer 1 Report

This research served the flowering phenology of 95 elepidote Rhododendron cultivars. The material, method, and result are acceptable for publication. I think it is useful information for marketing professionals, nursery growers, and plant breeders. However, the writing and discussion are not as good. I recommend some major editing in their academic writing, discussion method, and result presentation. 

Some suggestions in the following:

*Many paragrams have unknowing purposes. The first sentence doesn't show the topic well, the following sentences are not close or logically related. 

**Discussion is badly present (some examples in the following list). Please try to have a clear topic for each paragraph or discussion section. Ex, genetic (species) background in flower time, genetic (species) background in flower period, compare current data to earlier publications...

***Material description needed. 

Line 18-22, can almost be deleted. Only the last sentence is related information. 

Line 24-24, consider rewriting. 

Line 33-40, *

Line 108, *use week or data in the same discussion. Some sentences use week, and some use date is very messy. 

Line 139-246, are not closely relative to your research. This research isn't really based on the environment or soil. 

A table or figure that shows the flowering time a group is recommended

A supplement table of plant material (species name or possible group) is recommended

Line 303, a survey of flower phenology for 95 Elepidote Rhododendron cultivars.

Line 304-313 isn't a conclusion, just listed data. Maybe, flower time is correlated to the cultivar's pedigree (examples)

Line, This is not concluded from this research. A guessing market impact is not a conclusion. And, this research did not do any ornamental value survey. 

Practical implications are not related to your data. You can only recommend longer flower season cultivars from your data. 

Author Response

Thank you very much for all remarks. The paper is better now, than earlier.

Reviewer 2 Report

Flowering phenology is very important for ornamental plants because it can help us to select suitable taxa or varieties to extend ornamental period in one year. In this manuscript, authors selected 95 taxa of Elepidote Rhododendron including some common and lesser-known large flowered taxa.  

In this study, the different cultivars of shrubs were monitored for bud development. Subsequent stages were determined using a developed 10-degree comparative scale. The explicit flowering stages divided will help us to know the flowering periods of different cultivars in one year.  

From this study, we can get the conclusions, included the earliest flowering cultivars and the latest ones, the cultivars with the longest full bloom period, cultivars with particularly ornamental foliage, and so on. The results provid the basic information of ornamental values so that we can apply the common and lesser-known large flowered taxa.  Authors  recommended some cultivars, such as for compositions in shades of white and pink, for compositions in shades of red, for small space gardens, for small space gardens, in areas with harsh and cold winters, of the cultivars most commonly marketed. 

Based on the results of this study, we will have more selection in Rhododendron application.

By the way, author should list the information of Rhododendron  taxa selected in this study including the species or cultivar names, localities, ages, origin sources, breeders, and so on.  By the imformation table, we can jude the realationship, ecological habits, flowering characteristics of different cultivars.

Author Response

Thank you very much for all remarks. The table with leading species, parentage and breeder is added.

Round 2

Reviewer 1 Report

This paper provides information for gardeners and breeders about the flower time and flower period of 95  Rhododendron taxa. After being revised, I think this manuscript is ready to publish, just needs some small editing. 

Tables: Make sure the format is fit into the journal. Smaller space for each row and some editing to avoid the second line of each row would be preferred.